# Modulation of the Structural, Magnetic, and Dielectric Properties of YMnO_3_ by Cu Doping

**DOI:** 10.3390/ma17122929

**Published:** 2024-06-14

**Authors:** Feng Wan, Xuexia Hua, Qiufen Guo

**Affiliations:** College of Physics & Electronic Engineering, Xianyang Normal University, Xianyang 712000, China

**Keywords:** YMn_1-*x*_Cu*_x_*O_3_, structure, magnetic, dielectric

## Abstract

The lower valence compensation of YMn_1-*x*_Cu*_x_*O_3_ (*x* = 0.00, 0.05, and 0.10) is prepared by the solid-state reaction, and the effects of divalent cation Cu-doping on the construction and magnetic and dielectric attributes of multiferroic YMnO_3_ are systemically researched. Powder X-ray diffraction shows YMn_1-*x*_Cu*_x_*O_3_ has a single-phase hexagonal construction with a *P6_3_cm* space group as the parent YMnO_3_, and lattice parameters decrease systematically as Cu concentration increases. Using the scanning electric microscope, structure morphologies analysis shows that the mean grain size varies between 1.90 and 2.20 μm as Cu content increases. YMn_1-*x*_Cu*_x_*O_3_ magnetization increases as Cu doping concentration increases, and the antiferromagnetic transition temperature declines from 71 K for *x* = 0.00 to 58 K for *x* = 0.10. The valence distributions of Mn ions conduce to the modified magnetic attributes. Due to Cu substitution, the dielectric loss and dielectric constant decline as frequency increases from 400 to 700 K, showing representative relaxation behaviors. Indeed, that is a thermally activated process. In addition, the peak of the dielectric loss complies with the Arrhenius law. The relaxation correlates to the dipole effect regarding carrier hopping between Mn^3+^ and Mn^4+^, and also correlates to oxygen vacancies generated by Mn^2+^.

## 1. Introduction

Hexagonal rare-earth manganites, YMnO_3_ (*h*-YMnO_3_) as a typical example of a multiferroic material, have attracted considerable interest owing to their promising physical properties and remarkable applications in spintronics and memory devices [1,2,3,4,5,6,7]. *h*-YMnO_3_ exhibits a ferroelectric transformation at *T*_C_~900 K as well as an antiferromagnetic order below *T*_N_~70 K. *h*-YMnO_3_ crystal consists of MnO_5_ trigonal bipyramids, separated by Y^3+^ ions along the *c*-axis [8,9]. MnO_5_ bipyramids are formed by Mn^3+^ ions, two apical oxygen atoms (O1 and O2), and three in-plane oxygen atoms (O3 and O4). MnO5 trigonal bipyramids tilt to the *ab*-plane, leading to the manganese ion displacing from the bipyramid center, which causes the Mn^3+^ triangular network lattice to distort [10,11]. The antiferromagnetic ordering originates from the super-exchange interactions between Mn^3+^ in the *ab*-plane. Simultaneously, to maintain the stability of the structure, the bond lengths of the Y-O at the top are not equal, which leads to polarization and ferroelectricity. The ferroelectricity and magnetism originating from various sources result in substantial variations in the ferroelectric transition temperature and spin-ordering temperature. Moreover, the application possibilities of YMnO_3_ are limited by the weak coupling of electricity and magnetic fields, which makes this material much more attractive [1,3,12,13]. 

To enhance magnetoelectric coupling and comprehend the multiferroic mechanism in YMnO_3_, many studies have valued the impact of substitution on the Mn or Y sites. Elemental doping is among the top crucial and common methods of altering physical attributes. The change in the tilting of the MnO_5_ trigonal bipyramid can be tuned by doping ions with distinct radii and variable valences at the Y or Mn sites. Many studies have been conducted on dielectric behaviors, electrical properties, and magnetic properties of YMnO_3_ by replacing the elements like Sr, Ca, Dy, Lu, Gd, or Zr at the Y site, or Mn, Mn, Al, Co, Ti, etc., at the Mn site [10,11,13,14,15,16,17,18,19,20,21,22,23,24,25,26]. For instance, when doping with Lu at the YMnO_3_ Y site, the dielectric constant achieved the pinnacle within the solid solution area, but the Curie temperature did not alter [27]. As reported by Rajesh, divalent Sr^2+^ ions substitute Y^3+^ ions, which introduces the ferromagnetic (FM) duel exchange of Mn^3+^-O^2−^-Mn^4+^ and reduces the magnetic frustration ratio *f* = |*θ*_CW_|/*T*_N_. The influence of Sr doping on the magnetic properties of YMnO_3_ suggests the redirection of the Mn moment in the basal plane perpendicular to the initial orientation at ~38 K [20]. Dielectric research on Y_1-*x*_Dy*_x_*MnO_3_ illustrates that the dielectric constant increases significantly as Dy doping increases, and double ionization oxygen vacancy relaxation occurs at 510 K [21]. Dielectric spectroscopy analysis of Ba-doped YMnO_3_ by Jyoth et al. showed that the dissipation factor and dielectric constant decreased as frequency increased, which denoted that space charge polarization plays an important role [17]. 

According to the literature survey, doping studies on the YMnO_3_ have mostly concentrated on the Mn sites. By substituting Mn^3+^ with magnetic ions like Ru^3+^, Fe^3+^, Ni^3+^, Co^2+^, Os^2+^, etc., the magnetic properties have been investigated intensively [10,11,12,21,22,23,24,25,26]. Perhaps, a new state emerges from the frustrated state, on account of diverse perturbations within a two-dimensional triangular network through doping [27]. Thus, it is interesting to investigate the modification of the ferroelectric and antiferromagnetic properties and magnetoelectric coupling in YMnO_3_ compounds. Recently, a large exchange bias effect in Fe-doped YMnO_3_ films at 2 K has been reported. This exchange bias is attributed to the interaction of the antiferromagnetic and spin glass states at low temperatures [28]. Olivera reported that the leakage currents were decreased by the partial substitution of Mn^3+^ with Ti^4+^ in YMnO_3_; however, the ferroelectric response was not improved remarkably. The increased magnetization and weak ferromagnetism indicated that the antiferromagnetic ordering between Mn^3+^ ions was suppressed by the nonmagnetic Ti^4+^ in YMnO_3_ [12]. However, there have been few reports on the substitution of Cu^2+^ ions in YMnO_3_. The orthogonal YMn_2/3_Cu_1/3_O_3_ electronic structure studied by Aoskan et al. using synchronous radiation X-ray absorption showed that doping with bivalent Cu^2+^ increased the valence state of Mn^3+^ ions and simultaneously enhanced the non-localization of *e_g_* electrons of Mn 3*d* orbitals, resulting in improved conductivity in the sample [29]. As observed by Jeuvrey et al., Cu doping affects the magnetic attributes of YMnO_3_, and the antiferromagnetic transition temperature of Cu-doped YMnO_3_ decreased significantly [30]. However, Xiao et al. suggested that substituting Mn^3+^ ions with Cu^2+^ ions had little effect on the antiferromagnetic transition temperature [31]. So far, research on the structure of Cu-doped hexagonal YMnO_3_ is insufficient, and the mechanism of the hybridization of Mn and O ions and the impact of varying electronic structures on the magnetic attributes of YMnO_3_ are still unclear. Gutierrez’s research indicates that when Cu doping exceeds 20%, the sample exhibits a perovskite structure of YMnO_3_ [29]. In this work, YMnO_3_ with a doping amount of less than 10% of Cu ions was selected for investigation. The Cu concentration is high enough to show the effect of Cu doping on the structural, magnetic, and dielectric behaviors and can avoid any instability in the crystal structure. The Cu-doped h-YMnO_3_ ceramic was synthesized through the solid-state reaction method. The structural, magnetic, as well as dielectric behaviors are systematically studied. 

## 2. Experimental Details

Ceramic polycrystalline samples of YMn_1-_*_x_*Cu*_x_*O_3_ (*x* = 0.00, 0.05, and 0.10) were synthesized by a typical solid-state reaction technique using stoichiometric quantities of high-purity Y_2_O_3_, Mn_2_O_3,_ and CuO. The mixed raw materials were calcined in air at 1100 °C for 24 h by interval grinding. To improve the homogeneity, the precursor powders were compacted into tiny pellets under 10 tons of hydraulic pressure and 5% polyvinyl alcohol as an adhesive. The samples were heated up to 1035 °C and held for 12 h. The crystalline structures of all samples were characterized in the angular range from 10° to 80° with a step of 0.02° and scanning speed of 2°/min. through X-ray powder diffraction (XRD) with Cu Kα radiation (λ = 1.5460 Å). The XRD modes were validated through Rietveld analysis with the FullProf project. The surface topography of the samples was examined through scanning electron microscopy (SEM). Moreover, the distribution of grain size was gauged through Nano Measurer software (Nano Measurer 1.2). X-ray photoelectron spectroscopy (XPS) with Al Kα radiation was used to analyze the variation of ionic valence state for Cu-doped YMnO_3_ powder, and the obtained curves were fitted with XPS-PEAK4.1 software according to the Gauss–Lorentz line. The binding energy was corrected using the neutral carbon peak of C 1 s, which was assigned a value of 284.6 eV to compensate for surface charge effects. Magnetization was performed through a Superconducting Quantum Interference Device (SQUID) at 2 K-300 K, showing a magnetic field of H = 1 kOe in zero-field cooling (ZFC) and field-cooling (FC) patterns. Dielectric attributes were determined in the frequency of 1 kHz to 1 MHz at 300 K-900 K through an HP4284LCR meter.

## 3. Results and Discussion

The X-ray diffraction (XRD) patterns for synthesizing YMn_1-_*_x_*Cu*_x_*O_3_ (*x* = 0.00, 0.05, and 0.10) powders sintered at 1350 °C for 30 h are shown in Figure 1. As verified through JCPDS card no. 25-1079, the total samples exhibit a hexagonal structure showing the space group *P6_3_cm*. No other diffraction peaks of impurities are observed in the XRD patterns. It is noticeable that the diffraction peaks of the XRD pattern (*x* = 0.05 and 0.10) slightly shift to higher 2*θ* values compared with *x* = 0.00, which proves that Cu has been successfully doped into YMnO_3_. The (112) and (110) peaks of the *x* = 0.00 shift towards higher 2*θ* values are shown in Figure 1a. This suggests that the lattice parameter decreases with Cu doping. Rietveld refinement was implemented to obtain the explicit structural information of the total samples. As shown in Figure 1b, a schematic depiction of the atomic arrangements within the hexagonal lattice of YMnO_3_ exhibits a *P6_3_cm* space group. In the XRD structure refinement, the Y ions occupied two specific sites: Site 2a, located at (0, 0, z), and Site 4b, located at (1/3, 2/3, z). Simultaneously, the Mn/Cu ion was situated at Site 6c, surrounded by five oxygen ions, designated by (x, 0, z). The five oxygen ions at Sites 2a, 4b, and 6c have coordinates of (0, 0, z), (1/3, 2/3, z), and (x, 0, z), respectively [32]. Rietveld refinement for the samples was finished through Fullprof software (FullProf 2020.6), and the peaks were fitted with the Pseudo–Voigt function. Refined patterns are shown in Figure 1c. In addition, refinement parameters, such as *R_wp_*, *R_p_*, and *R_exp_*, were considered in terms of refinement course. Refined parameters are displayed in Table 1. The ionic radius of Cu^2+^ is 0.73 Å which is larger than Mn^3+^ (0.65 Å), Mn^4+^ (0.53 Å), and Mn^2+^ (0.67 Å). The decrease in lattice parameters with increasing Cu^2+^ amount, as shown in Table 1, suggested that partial substitution of Cu^2+^ ions with Mn^3+^ in YMnO_3_ will not only result in a change in the blended-valence state of Mn ions, but also cause a chemical stress effect. Compared with parent YMnO_3_, most Mn-O bond lengths increased with increasing Cu concentration. The Y-O bond length of parent YMnO_3_ was longer than that of Cu-doped YMnO_3_, which illustrates that Cu doping raises Mn trimerization in the *ab*-plane and decreases the *c* structure parameter [31]. The tolerance factor, which determined the stability of prepared perovskite ceramic, increased from 0.891 for x = 0.00 to 0.893 for x = 0.10. The addition of Cu ions can cause a decrease in the average ionic radius of Mn sites, which is the main reason for the increase in the tolerance factor. This leads to the formation of ferromagnetic properties in YMnO_3_ [33]. This validates that a stable perovskite structure can be formed for all prepared ceramic samples. Thus, the increase in the Cu^2+^ ions with *x* may exhibit that the distortion of the samples intensifies. 

In view of the isotropic essence of the crystal, the microstrain and average crystallite size are achieved through the Williamson–Hall equation:(1)$$\beta_{hkl}\cos\theta =\frac{K\lambda }{D_{v}}+4\varepsilon \sin\theta$$
where *D_v_* means the volume-weighted crystallite size, *K* means the shape factor, and *ε* means lattice strain. As exhibited in Figure 2, a graph is drawn through 4sin*θ* along the X-axis and *β_hkl_*cos*θ* along the Y-axis for the samples (*x* = 0.00, 0.05, and 0.1). The lattice strain and crystallite size were derived from a linear fit. The estimation values of the strain and size are depicted in Table 1. The crystallite size declines as Cu content increases. At the same time, the lattice strain varies between 6.03 × 10^−4^ for *x* = 0.00 and 8.73 × 10^−4^ for *x* = 0.10.

Figure 3 shows the scanning electron micrograph (SEM) of the morphology of YMn_1-*x*_Cu*_x_*O_3_ powders annealed at 1350 °C and the distribution histogram of the particle size through Nano Measurer software. Figure 3a shows larger particles and cracks in the *x* = 0.00 sample. The generation of cracks may be related to the heat treatment temperature. As shown in Figure 3b,c, as the amount of Cu doping increases, the sample particles gradually decrease and most of the uniform particles are mutually necked. From the histogram corresponding to SEM, it can be seen that the grain size of 90% of the particles in each sample varies in the range of approximately 1–4 μm. For *x* = 0.00, most particles were distributed between 2 and 2.5 μm in the sample. With increasing Cu doping, particles of 1.2 to 2 μm appeared in the samples. Thus, the mean grain size decreases from 2.14 μm for undoped YMnO_3_ to 1.88 μm for *x* = 0.10, which corresponds to the results through the Williamson–Hall equation fitting in XRD. However, all sample grain sizes highly exceed the crystallite sizes estimated by the Williamson–Hall equation. This is because the grain size by XRD analysis is the coherent domain of microcrystals, whereas the average grain analysis by SEM is formulated by aggregating multiple crystallites in the sintering procedure [10,15].

X-ray photoelectron spectroscopy (XPS) has become an efficient tool for studying valence-change ions [22]. Figure 4 shows the XPS spectra of the Mn 2*p* and O 1*s* regions of YMn_1-*x*_Cu*_x_*O_3_ (*x* = 0.00, 0.05, and 0.10) at ordinary temperature. The Mn 2*p* peak is divided into 2*p*_3/2_ and 2*p*_1/2_, on account of spin–orbit coupling. The Mn 2*p*_3/2_ peak can be deconvoluted into differentiated peaks through Gaussian–Lorentzian curve fitting, which indicates that there are various valence Mn ions in the samples. The peaks at a binding energy of 641.28 and 643.67 eV are consistent with Mn^3+^ and Mn^4+^ oxidation states for *x* = 0.00, as shown in Figure 4a. In light of the fitting outcomes, the Mn^3+^:Mn^4+^ ratio for the YMnO_3_ ceramic is 75:25. This is close to the values reported in the literature [22,34]. The energy corresponding to 639.12 eV is the peak of the Mn^2+^ oxidation state, which is detected in the samples of *x* = 0.05 and 0.10. By calculating the peak region, the proportions of Mn^2+^:Mn^3+^:Mn^4+^ are 14:56:30 and 10:47:43 for *x* = 0.05 and 0.10, respectively, which illustrates that the trivalent state predominates in the mixed-valence manganese ions. The Mn^4+^ concentration increases as Cu doping increases. On account of the charge compensation, the introduction of Cu^2+^ (3*d*^9^) into the YMnO_3_ system transforms a portion of Mn^3+^ (3*d*^4^) into Mn^4+^ (3*d*^3^). The existence of Mn^2+^ states is associated with oxygen deficiencies caused by doping or the sample fabrication process.

The O1s spectrum revealed the hybridization of O2p with Mn 3d ions. The oxygen vacancy concentration changes with Cu content, which is validated by XPS analysis. Figure 4d–f exhibit the O1s core-level XPS spectra of Cu-doped YMnO_3_ with x = 0.00, 0.05, and 0.10. The main peak at approximately 529.50 eV, labeled as O1s, is attributed to the lattice oxygen in its normal position, with a reported range of 528–530 eV. [22]. Close to the O1s peak, another peak around 531.5 eV is ascribed to oxygen vacancies originating from the heat treatment or lattice distortion in the YMnO_3_ structure due to Cu doping [34]. The high-energy peaks located at the banding energy of 534.5 eV are assigned to the chemisorption of oxygen or adsorbed oxygen species from OH on the surface. XPS analysis results show that the relatively large contribution of the peak corresponds to the banding energy at 531.5 eV as Cu doping concentration increases, which denotes fewer oxygen vacancies in YMn_1-x_Cu_x_O_3_ series samples. This is most probably attributed to lattice distortion within the YMnO_3_ structure, which was induced by the incorporation of Cu. These oxygen vacancies are expected to produce free carriers (electrons), which will contribute to increasing the exchange interaction effects between magnetic impurities.

Figure 5 shows magnetization versus temperature curves at 1000 Oe for YMn_1-*x*_Cu*_x_*O_3_ (*x* = 0.00, 0.05, and 0.10) with increasing temperature between 5 K and 300 K in zero-field-cooled (ZFC) and field-cooled (FC) patterns. As depicted in Figure 5a, the ZFC and FC magnetization for YMnO_3_ rises as temperature declines and there exists a divergence between FC and ZFC data below *T*_N_~71 K. By comparison with the ZFC curve, the FC data rose slightly, which should be ascribed to a weak ferromagnetic component in the YMnO_3_ ceramics [26], which aligns with former reports on magnetic measurements of YMnO_3_ [23,26]. A similar change in ZFC-FC magnetization is also examined in *x* = 0.05 and 0.10, while the transition temperature *T*_N_ decreased to 58 K, rapidly. The antiferromagnetic transition temperature *T*_N_ decreased from 71 K to 58 K, which is consistent with the findings reported by Xiao et al. [26]. Moreover, as the amount of Cu doping was increased, the magnetization intensity increased over the entire temperature range. This notable difference at low temperatures implies an increase in the size of ferromagnetic clusters in the Cu-substituted sample. According to the charge compensation, introducing Cu^2+^ at the Mn site will generate Mn^4+^ in the sample. Based on the XPS analysis of the O element, oxygen vacancies gradually decreased with an increase in the doping amount. The origin of Mn^2+^ ions in the YMnO_3_ ceramics is due to the presence of oxygen vacancies, which is common in perovskite oxides. Three of the four electrons in Mn^3+^ (3*d*^4^) occupy the *t*_2g_ energy level, while the remaining one enters the *e*_g_ level. For Mn^4+^ (3*d*^3^) ions, three electrons are in the *t*_2g_ orbital. Since for Mn^2+^ (3*d*^5^), there is no crystal field splitting, instead of an *e*_g_ and *t*_2g_ orbital, now there is a *d*^5^ orbital with five electrons. Thus, the exchange interactions *e*_g_-*t*_2g_ (Mn^3+^-O^2−^-Mn^4+^), *e*_g_-*d*^5^ (Mn^3+^-O^2−^-Mn^2+^), and *t*_2g_-*d*^5^ (Mn^4+^-O^2−^-Mn^2+^) exist in the Cu-doped sample. At low doping levels, Hunds coupling is strong and the *e*_g_ spin is determined by both *e*_g_-*t*_2g_ and *e*_g_-*d*^5^ interactions. XPS analysis revealed that the concentration of Mn^3+^ ions in the sample decreased, as the Cu content increased. This result indicates that as the Cu-doping content increases, and the ferromagnetic interactions between Mn^3+^-O^2−^-Mn^4+^ and Mn^3+^-O^2−^-Mn^2+^ in the sample increase. Then, the magnetization of the sample increases with the increase in the doping amount of Cu^2+^ ions.

Figure 6 shows the inverse susceptibility (χ^−1^) versus T at a magnetic field of 1000 Oe for the samples in the FC mode. The χ^−1^(T) curve is capable of being fitted through the Curie–Weiss law above 150 K (paramagnetic state).
(2)$$\chi =C/\left(T-\theta_{CW}\right)$$
where *C* refers to the Cure constant and *θ*_CW_ refers to the Cure–Weiss temperature. 

The lines represent the Curie–Weiss fit of the results. Using Curie constant values, the magnetic moment of each Mn atom (*μ_eff_*) is computed as below:(3)$$\mu_{eff}=\sqrt{\frac{3k_{B}C}{N_{A}\mu_{B}^{2}}}$$
in the above equation, *x*, *N,* and *k*_B_ mean the Cu atom concentration, the Avogadro quantity, and the Boltzmann constant, respectively.

The magnetic features are listed in Table 2. The total samples possess negative *θ*_CW_, which denotes the antiferromagnetic interactions are still predominant in Cu-doped YMnO_3_ ceramics. However, it was discovered that the |*θ*_CW_| decreases from 456.714 K for *x* = 0.00 to 181.509 K with 10%-doped Cu^2+^ ions. This result indicates that fewer Mn^3+^ ions and Mn^3+^-O^2−^-Mn^3+^ antiferromagnetic super-exchange interactions in the triangular network were suppressed by Cu^2+^ doping. The apparent improvements in Mn^3+^-O^2−^-Mn^4+^ and Mn^3+^-O^2−^-Mn^2+^ ferromagnetic double-exchange are conducive to the improved weak ferromagnetic attributes with increasing concentration of Cu, even though weaker Cu^2+^-O^2−^-Mn^3+^ and Cu^2+^-O^2−^-Mn^4+^ antiferromagnetism are also introduced in Cu-doped samples.

The experimental values of the magnetic moment per Mn atom decreased from 5.249 μ_B_ to 4.859 μ_B_ with increasing Cu content. The decrease in *μ*_eff_ value in the experiment is consistent with that of the theoretical results. This decrease might have resulted from the introduction of electrons by Cu^3+^ (3*d*^9^). The effective magnetic moment of Cu^2+^ is 1.7 μ_B_ smaller than that of Mn ion. On the other hand, the substitution of Mn^3+^ ions by divalent Cu^2+^ ions introduces the Mn^4+^ ions to the crystalline structure according to the charge compensation theory. The XPS analysis exhibits Mn^3+^ and Mn^4+^ in Cu-doped YMnO_3_ compounds, showing that Mn^4+^ increased with doping, which is in agreement with the theory. The Y ions did not contribute magnetically to the system. Substitution of Cu^2+^ at the Mn^3+^ site results in one electron of the Mn^3+^ (3*d*^4^) ion in YMn_1-*x*_Cu*_x_*O_3_ being captured by Cu^2+^ (3*d*^9^) to form a more stable Cu^+^ (3*d*^10^). The Mn^4+^ ‘carries’ a hole that can hop in the Mn ions by hybridizing Mn *d*-states with the O *p*-states [11]. The Mn^3+^ component simultaneously transforms into Mn^4+^. The efficient magnetic moments of Mn^3+^ (3*d*^4^) and Mn^4+^ (3*d*^3^) reach 4.9 and 3.87 μ_B_, respectively. The theory’s effective magnetic moment (*µ_eff_*) was 4.900, 4.741, and 4.580 μ_B_ for YMn_1-*x*_Cu*_x_*O_3_ with *x* = 0.00, 0.05, and 0.10, respectively. For pure YMnO_3_, the experimental effective magnetic moment is smaller than the theoretical value mainly because of the formation of holes at the oxygen sites during the sintering process, captured by Mn^3+^ ions to form Mn^4+^ ions. The XPS results also confirmed the presence of Mn^4+^ ions. This result is consistent with those of other studies [18,22,34]. However, the experimental effective magnetic moment of Cu-doped YMnO_3_ exceeds the theoretical value, predominantly owing to the Mn^2+^ of the sample.

Chemical substitution is a key influencing factor of geometrical frustration. It is widely known that the inter-plane coupling neglects YMnO_3_, since the closet adjacent inter-plane coupling contributes less to 3 × 3 structures (the distance between neighboring Mn atoms reaches 3a), as exhibited in Figure 7a. As a measurement of geometrical frustration of the antiferromagnetic system, the frustration parameter (*f*) is regarded as the proportion of the Curie–Weiss constant to the Néel temperature:(4)$$f=\left|\theta_{CW}\right|/T_{N}$$

It reduces from 6.432 for *x* = 0.00 to 3.129 for *x* = 0.10., illustrating that Cu doping reduces the geometrical frustration of the triangular lattice. Accordingly, stringent antiferromagnetic ordering occurs.

The Mn ion in hexagonal YMnO_3_ is located at the center of the MnO_5_ triangular double pyramid. As exhibited in Figure 7b, the 3*d* orbitals of Mn ions are divided into two doublets, *e*_1_*_g_* (*yz*_↓_*/zx*_↓_) and *e*_2_*_g_* (*xy*↓/*x*^2^-*y*^2^_↓_), and a singlet, *a*_1_*_g_* (*z*^2^↓). The inter-plane AFM super-exchange interaction (*J*_nn_) between Mn-O3-Mn and Mn-O4-Mn is formulated through the overlap between Mn 3*d*_(x2-y2)_ or *d*_xy_ and O(3, 4) 2*p*_xy_. The Mn-O1-O2-Mn AFM super-exchange interplay takes place through the overlap between the O(1, 2) 2*p*_z_ orbital tails [35,36]. Rietveld results show that the bound distance between Mn and O along the *c*-axis is larger than that in the *ab*-plane. Thus, we consider the *ab*-plane to have a major impact. The bond length of Mn-O3 decreases with the increase in Cu doping. Due to orbital hybridization between Mn and O3, this result indicates a decrease in the number of electrons in the *e*_2_*_g_* (*xy*↓/*x*^2^-*y*^2^_↓_) orbital. Thus the Coulomb interaction between Mn and O3 is weakened. Furthermore, the inter-plane AFM super-exchange interaction weakened with increasing Cu. The *T_N_* are 71, 65, and 58 K for *x* = 0.00, 0.05, and 0.10, respectively, which can be ascribed to the decline in antiferromagnetic exchange interplay. Meanwhile, magnetization is enhanced because the Mn^3+^ transforms into Mn^4+^ ions.

According to the XXZ triangular antiferromagnetic model, the magnetic transition temperature is computed as follows:(5)$$T_{N}=-tJ_{\mathrm{nn}}(S+1/2)$$
where *t* aligns with the important temperature, and *J*_nn_ and *S* can be achieved from the following equations:(6)$$\theta_{CW}=\frac{1}{3}ZJ_{\mathrm{nn}}S\left(S+1\right)$$
(7)$$S=\sqrt{\left(1-x-y\right){\left(S_{Mn3+}\right)}^{2}+x{\left(S_{Mn4+}\right)}^{2}+y{\left(S_{\mathrm{Mn}2+}\right)}^{2}}$$
where Z = 6 stands for the quantity of the closest neighbors, and *S* is equal to 3/2, 2, and 5/2 for Mn^4+^, Mn^3+^, and Mn^2+^, respectively. Additionally, *x* and *y* stand for the ratios of Mn^4+^ and Mn^2+^, respectively, and they are determined by the area under the XPS binding energy curve.

The exchange integral *J_nn_* was −38.807, −26.298, and −16.414 for *x* = 0.00, 0.05, and 0.01. A negative value indicates that the antiferromagnetic interaction is predominant. The decreased value indicates a competition between ferromagnetic and antiferromagnetic interactions, owing to antiferromagnetic exchange Mn^3+^-O^2−^-Mn^3+^ interactions which are strong yet have a long range, whereas ferromagnetic exchange Mn^3+^-O^2−^-Mn^4+^ and Mn^3+^-O^2−^-Cu^2+^ interactions are strong yet have a short range [23]. The experiment shows that the *t* value increases from 0.731 to 1.469, indicating that copper substitution favors spin alignment in the *ab*-plane.

To explore the influence of Cu^2+^ on the dielectric behavior of YMnO_3_, dielectric measurements were implemented from 300 K to 900 K. Figure 8 shows the temperature dependence of dielectric constant and loss (*x* = 0.05 and *x* = 0.10) samples determined at discrepant frequencies from 1 KHz to 1 MHz, respectively. The overall dielectric constant values of *x* = 0.10 are higher than those of the *x* = 0.05. The dielectric constant and dielectric loss decrease as frequency rises at the identical temperature. An obvious dielectric relaxation effect can be examined from 400 K to 700 K; in addition, the dielectric relaxation peak shifts to more elevated temperatures as frequency rises. This result is consistent with the findings of Ma et al. on the dielectric relaxation of YMnO_3_, as well as other researchers’ findings on the impact of Co and Sr doping on dielectric properties [16,20,37]. To expand a deeper awareness of the dielectric relaxation of Cu-doped YMnO_3_, the activation energy *E*_a_ was calculated according to the Arrhenius law:(8)$$f=f_{0}\exp(-E_{a}/k_{B}T)$$
where *f*_0_ stands for the characteristic relaxation frequency at limited temperatures, *E*_a_ stands for the activation energy, *k*_B_ stands for the Boltzmann parameter, and *T* stands for the peak temperature.

Figure 9 exhibits the peak temperature as a function of frequency, and experimental data are greatly fitted to the Arrhenius law. More specifically, the activation energies (x = 0.05 and x = 0.10) reach 0.76 eV and 1.23 eV, respectively. A rise in doping concentration triggers various compensatory mechanisms. As reported by Moure et al., 0.36 eV is achieved through DC conductivity measurements for the temperature of 330 to 500 K, and that conduction mechanism is thermally activated hopping of small polarons between local positions of Mn^3+^ and Mn^4+^ [38]. The oxide vacancy is an important factor for the activation energy is 0.76 eV between 450 and 720 K for Cr-doped YMnO_3_ [10]. Generally, ionic valence significantly affects the dielectric attributes of the material. For Cu-doped samples, the XPS outcomes indicate that Mn^2+^, Mn^3+^, and Mn^4+^ emerge simultaneously in Cu-doped samples, and that Mn^4+^ increases as Cu doping increases. The valence from Mn^3+^ to Mn^4+^ occurs, on account of the hole conduction mechanism. This suggests that Cu^2+^ might trigger an increased transition from Mn^3+^ to Mn^4+^ ions, and increase charge hopping. Therefore, the dielectric constant and activation energy emerge.

The activation energy of Cu-doped YMnO_3_ is much larger than that of Cr-doped samples reported in the literature, which may be mainly related to other factors. Generally, an increase in doping concentration induces a range of compensation mechanisms, such as electrical, B-site, and oxide vacancy compensation. These mechanisms can result in significant polarization under an applied electric field. XPS analysis showed that the content of oxygen vacancies in the sample decreased from 41% for x = 0.05 to 10% for x = 0.10. The presence of oxygen vacancies primarily results from lost traces of oxygen during sintering at higher temperatures, and electrons are created. It may be represented by the following equation:(9)$$V_{O}^{\times }\to O_{O}^{••}+2e^{'}+\frac{1}{2}O_{2}$$

The reduction in oxygen vacancies results in decreased ionic conductivity and increased activation energy, which weakens the dielectric response. Meanwhile, the generated electrons are captured by Mn^3+^ to form Mn^2+^.
(10)$${\mathrm{Mn}}^{3+}+e^{\prime }\to {\mathrm{Mn}}^{2+}$$

The ordering of Mn^2+^ and Mn^3+^ can produce the polar clusters and the thermally activated dielectric relaxation will be intensified. Thus, the primary factors responsible for the increase in dielectric constant and dielectric relaxation, as well as the relaxation activation energy increasing with Cu doping, are not only the strengthened dipole effect between Mn^3+^, Mn^2+^, and Mn^4+^ due to Cu doping, but also the decreased oxygen vacancies, which is a significant mechanism contributing to this outcome.

## 4. Conclusions

Hexagonal YMn_1-*x*_Cu*_x_*O_3_ ceramics were compounded using a solid-state reaction. The magnetic and dielectric properties of YMnO_3_ varied with Cu doping. X-ray diffraction confirmed the formation of a single hexagonal structure in YMn_1-*x*_Cu*_x_*O_3_. An increase in the weak ferromagnetic component was detected in the Cu-doped sample. The Curie–Weiss temperature decreased because of the weakened AFM exchange interaction of Mn^3+^-O^2−^-Mn^3+^ with Cu^2+^. The dielectric activation energies enhance with Cu concentration, indicating that the Cu-doped YMnO_3_ samples are controlled by the hopping charge carriers between Mn^3+^, Mn^2+,^ and Mn^4+^. Oxygen vacancies are another important factor that causes an increase in the activation energy and dielectric constant with doping.

## Figures and Tables

**Figure 1 materials-17-02929-f001:**
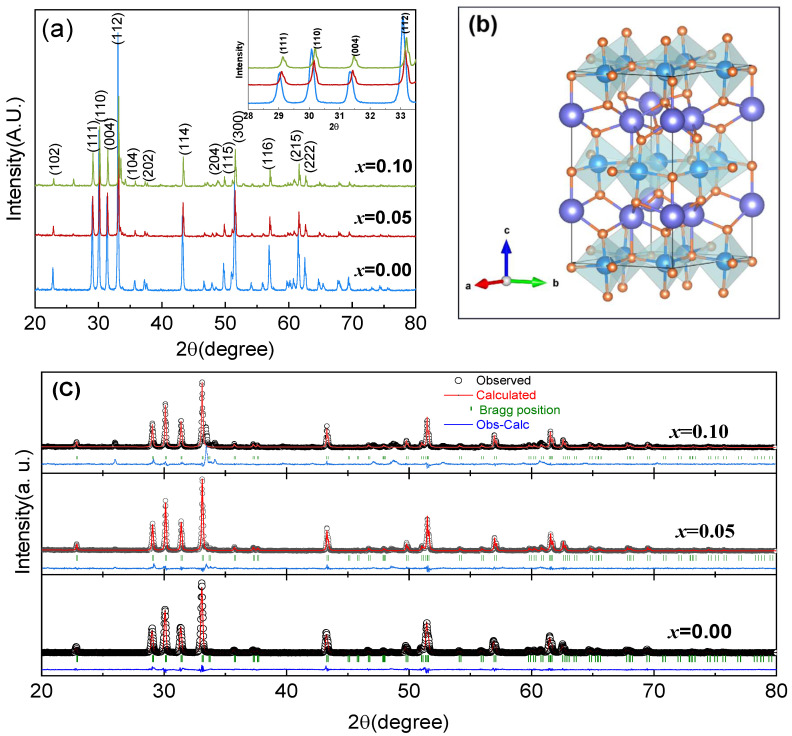
(**a**) Room-temperature XRD patterns for YMn_1-*x*_Cu*_x_*O_3_ (*x* = 0.00, 0.05 and 0.10). (**b**) Schematic structure for hexagonal YMnO_3_. (**c**) Rietveld refinement plot for YMn_1-*x*_Cu*_x_*O_3_ (*x* = 0.00, 0.05, and 0.10).

**Figure 2 materials-17-02929-f002:**
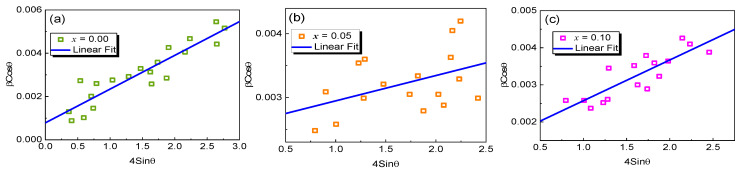
Williamson–Hall (W-H) plot of YMn_1-*x*_Cu*_x_*O_3_ ((**a**) *x* = 0.00, (**b**) 0.05, and (**c**) 0.10) ceramics.

**Figure 3 materials-17-02929-f003:**
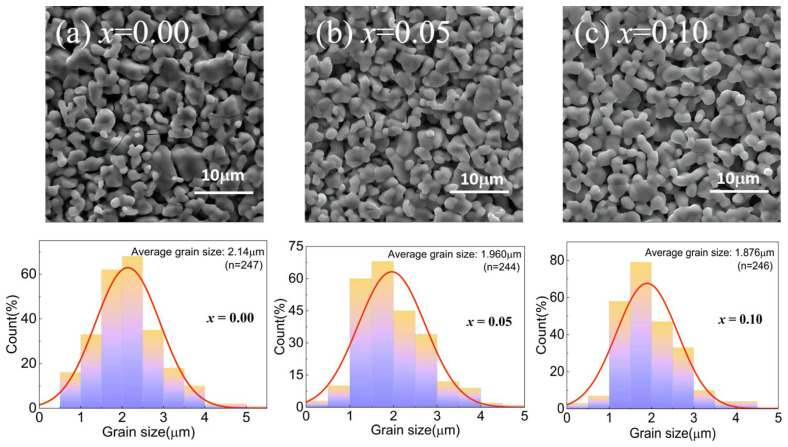
SEM micrographs of YMn_1-*x*_Cu*_x_*O_3_ ((**a**) *x* = 0.00, (**b**) 0.05, and (**c**) 0.10) with the grain size histograms.

**Figure 4 materials-17-02929-f004:**
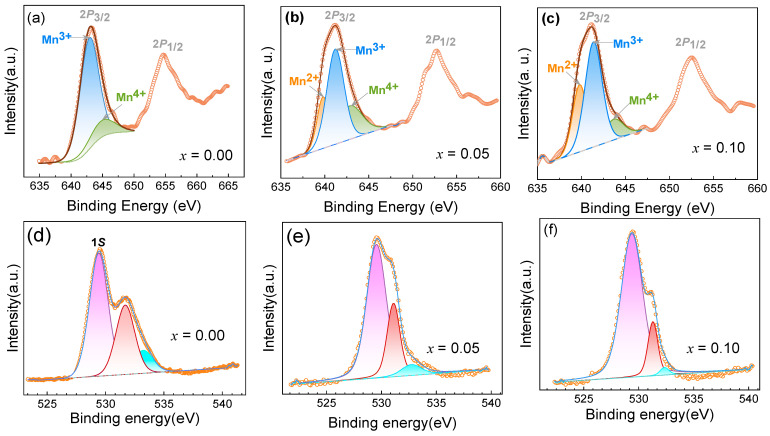
XPS spectra of Mn-2*p* and O-1*s* for YMn_1-*x*_Cu*_x_*O_3_ ((**a**,**d**) *x* = 0.00, (**b**,**e**) 0.05, and (**c**,**f**) 0.10).

**Figure 5 materials-17-02929-f005:**
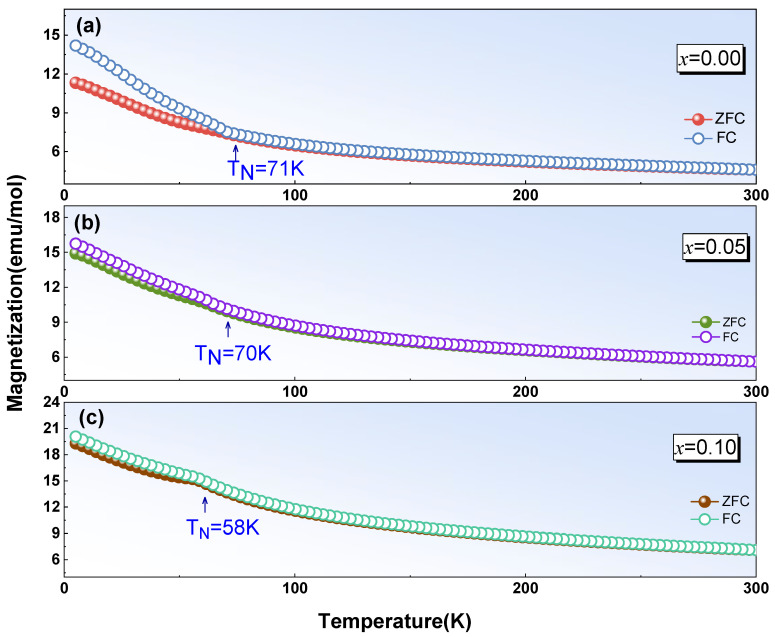
Temperature variation of magnetization with temperature for YMn_1−*x*_Cu*_x_*O_3_.

**Figure 6 materials-17-02929-f006:**
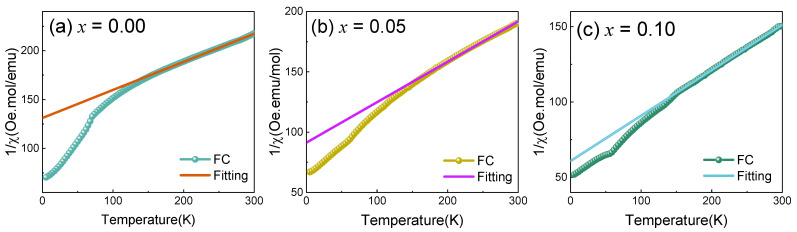
The temperature dependency of the FC inverse magnetic susceptibility for YMn_1−*x*_Cu*_x_*O_3._ The straight lines represent the fitting curves through the Curie–Weiss law.

**Figure 7 materials-17-02929-f007:**
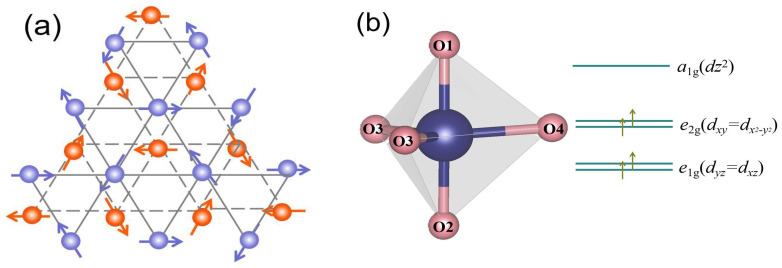
(**a**) The *ab*-plane of YMnO_3_ and the construction of Mn ions. Blue and red balls are integral to trimmers in *c* = 0 and *c* = 1/2, respectively. (**b**) The 3*d* orbital of Mn ions.

**Figure 8 materials-17-02929-f008:**
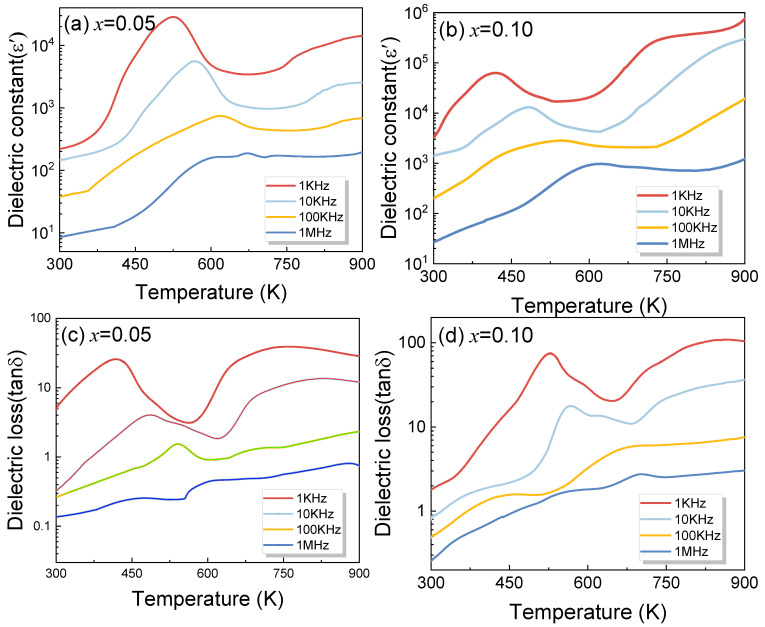
Temperature dependence of (**a**,**b**) dielectric constant and (**c**,**d**) dielectric loss under the frequencies between 1 kHz and 1 MHz for YMn_1-*x*_Cu*_x_*O_3_ (*x* = 0.05 and *x* = 0.10).

**Figure 9 materials-17-02929-f009:**
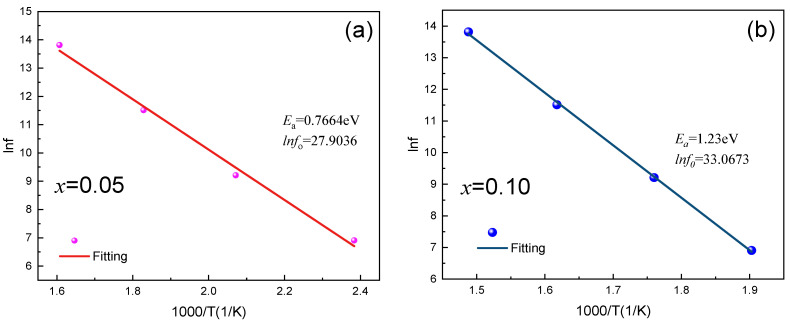
Frequency dependence of dielectric constant inflection point temperature for dielectric relaxation of YMn_1-*x*_Cu*_x_*O_3_ ((**a**) *x* = 0.05 and (**b**) *x* = 0.10) ceramics. The solid line represents the fitting curves through Arrhenius's law.

**Table 1 materials-17-02929-t001:** Structural parameters, selected bond lengths, and bond angles of YMn_1-*x*_Cu*_x_*O_3_ (*x* = 0.00, 0.05, and 0.10) ceramics obtained from Rietveld XRD refinement.

Sample	*x* = 0.00	*x* = 0.05	*x* = 0.10
a (Å)	6.15043	6.15292	6.15859
*c* (Å)	11.41665	11.39213	11.38779
V (Å)	374.007	370.235	374.052
*R*_wp_ (%)	21.3	7.81	12.6
*R*_p_ (%)	14.3	5.64	7.57
*R_ex_*_p_ (%)	5.22	5.32	5.19
χ^2^	2.02	2.15	5.88
Y1-O1 (Å)	2.26215	2.15725	2.18806
Y1-O2 (Å)	2.31963	2.19471	2.17532
Y1-O3 (Å)	2.36436	2.32077	2.25759
Y2-O1 (Å)	2.31216	2.40108	2.43321
Y2-O2 (Å)	2.32476	2.34721	2.30083
Y2-O4 (Å)	2.41465	2.56255	2.65046
Mn-O1 (Å)	1.82675	1.80449	1.79189
Mn-O2 (Å)	1.86583	1.99153	2.00210
Mn-O3 (Å)	2.07116	2.09712	2.09743
Mn-O4 (Å)	2.05808	2.05543	2.05234
Tolerance factor	0.891	0.892	0.893
Crystallite size from W-H plot (nm)	52.183	50.154	49.686
Micro strain	6.03 × 10^−4^	7.28 × 10^−4^	8.73 × 10^−4^

**Table 2 materials-17-02929-t002:** Magnetic parameters through Curie–Weiss law fitting of χ-T for the total compounds.

Sample	*T*_N_ (K)	C	*q*_CW_ (K)	*f* = |*q*_CW_|/*T*_N_	*µ*_eff_ (exp) (μ_B_)	*µ*_eff_ (calc) (μ_B_)
*x* = 0.00	71	3.483	−456.684	6.432	5.249	4.900
*x* = 0.05	65	3.321	−303.222	4.665	5.125	4.741
*x* = 0.10	58	2.984	−181.509	3.129	4.859	4.580

## Data Availability

The original contributions presented in the study are included in the article, further inquiries can be directed to the corresponding authors.

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
