# Peer review of "Modulation of the Structural, Magnetic, and Dielectric Properties of YMnO_3_ by Cu Doping"

_materials, 2024, doi:10.3390/ma17122929_

Round 1

Reviewer 1 Report

Comments and Suggestions for Authors

The article deals with the effects of Cu doping on the structural, magnetic, and dielectric properties compounds. Using various experimental techniques, the authors synthesized polycrystalline samples and conducted comprehensive analyses through X-ray diffraction (XRD), scanning electron microscopy (SEM), X-ray photoemission spectroscopy (XPS), magnetization measurements, and dielectric property assessments. The article revealed that Cu doping alters the lattice parameters, enhances weak ferromagnetic properties, decreases the Néel temperature and influences the dielectric behavior through charge carrier hopping and oxygen vacancy mechanisms.

However, some aspects could be improved:

· There is missing the detailed experimental findings, particularly in understanding the electronic structure changes due to Cu doping.

· The doping levels are limited to x = 0.00, 0.05, and 0.10 (missing explanation why).

· Authors should explain the fitting process and the significance of peak related to Mn oxidation states (deconvolution of Mn 2p).

· There is the huge number of abbreviations without explanations (similarly, some quantities are without any explanation).

· Add units for variables (in text or after expression definition).

· Some graphs (figures) are clear, but some of them are blurred and it is difficult to read the text (or data).

· Unify the references according to template.

· Unify the text and equations according to template (different fonts, types, and difficult to identify, what is function, what is variable or index).

Questions for the authors:

1. How does the presence of Mn2+ ions influence the overall magnetic and dielectric behavior of the doped samples?

2. How decrease in the tolerance factor with increasing Cu content influences the structural stability and potential applications of the material?

3. How did authors ensure the consistency and reliability of data across using of different methods (such as XRD, SEM, and XPS)?

Author Response

Dear Reviewer:

We thank you for giving us the opportunity to revise and resubmit our manuscript materials-3032889 entitled “Modulation of the structure, magnetic and dielectric properties of YMnO3 by Cu doping”. We are thankful to you for pointing out some important modifications. We have thoughtfully taken into account these comments. A thorough revision is in blue throughout the manuscript to reflect the improvement of scientific accuracy and presentation clarity. We hope that all these changes fulfil the requirements to make the manuscript acceptable for publication in materials

Our collective point-by-point responses to reviewers’ comments are provided in in the following pages. My co-authors and I want to thank you again for giving us the opportunity to revise and resubmit our manuscript to materials

Reviewer 2 Report

Comments and Suggestions for Authors

The manuscript explores how adding copper (Cu) affects the structure, magnetic properties, and dielectric behavior of YMnO₃, a multiferroic material. The researchers created samples of YMn₁-xCuxO₃ with different levels of Cu (x = 0.00, 0.05, and 0.10) using a solid-state reaction. They found that Cu doping leads to smaller lattice parameters, increased magnetization, and a lower antiferromagnetic transition temperature (TN). Using various techniques like X-ray diffraction, scanning electron microscopy, X-ray photoemission spectroscopy, and magnetometry, the study shows that the dielectric constant and loss decrease with frequency, showing a relaxation pattern that follows the Arrhenius law. These results help us understand how Cu doping changes the properties of YMnO₃, which could be useful for developing new technologies in spintronics and memory devices.

While the work looks interesting and contributes to the field, the following points need to be addressed: 

1. Improve the introduction. Make sure the introduction thoroughly covers the current research on Cu-doped YMnO₃ and similar materials. As well, clearly state the main research question or hypothesis of your study. This helps readers understand the purpose and significance of your work.

2. Provide more details about the experimental conditions, such as how temperature was controlled during synthesis and measurements.

3.Enhance the quality of your figures and tables. Ensure that all axes in the graphs are clearly labeled with units and that the resolution of images is high. At least in my copy they are not good enough. 

4. Discuss any potential sources of error in your experimental methods and how you addressed them. This transparency will add to the credibility of your results. In particular, the Figure 2 the linear fit seems to be trivial. Add the error bars.

5. The synthesis involves calcination at 1100°C for 24 hours followed by sintering at 1350°C for 30 hours​​. This high temperature and prolonged duration can lead to grain growth and potential inhomogeneity in the sample.

6. The powders are compacted into small pellets and reheated, but the homogeneity of this process and the uniformity of the applied pressure are not detailed​​. Also, variations in pressure could lead to inconsistencies in the final product.

7. The grain size distribution is measured using Nano Measurer software, but the sample preparation method for SEM is not described properly​​. Poor sample preparation can affect the accuracy of the grain size measurements.

8. XPS data is fitted using XPS-PEAK4.1 software with a Gauss-Lorentz line shape​​. The choice of fitting parameters and their justification are not discussed, which can affect the interpretation of oxidation states. In addition, XPS is surface-sensitive, and surface contamination or oxidation can affect the results. The manuscript does not mention in detail any surface cleaning procedures before XPS analysis​​.

9. Magnetization measurements are performed at 1 kOe using zero-field-cooled (ZFC) and field-cooled (FC) protocols​​. The temperature ramp rates and equilibration times at each temperature must be specified properly, which can impact the accuracy of the magnetization data.

10. Dielectric properties are measured from 1 kHz to 1 MHz at temperatures ranging from 300K to 900K​​. The stability and accuracy of the measurement setup at high temperatures are not discussed. High-temperature measurements can be affected by thermal drift and electrode polarization effects.

11. Larger particles and cracks are seen in the undoped sample, while Cu-doped samples show more uniform particles​​. The exact cause of these cracks isn't discussed, which could be related to the synthesis process or sample handling.

12. The paper reports a significant decrease in TN with Cu doping but doesn't compare these findings with existing literature or discuss possible reasons for differences with other studies.

13. The negative Curie-Weiss temperatures (θCW) show predominant antiferromagnetic interactions, but the decrease in |θCW| with increasing Cu doping suggests a reduction in antiferromagnetic exchange interactions​​. The paper doesn't fully explore the implications of this reduction on the overall magnetic behavior.

14. The dielectric constant and loss decrease with increasing frequency, showing relaxation behavior consistent with the Arrhenius law​​. However, the discussion lacks a detailed explanation of how Cu doping specifically affects the dielectric relaxation mechanisms.

15. The activation energies for dielectric relaxation are reported to be 0.76 eV and 1.23 eV for x = 0.05 and x = 0.10, respectively​​. The paper doesn't thoroughly discuss why the activation energy increases with higher Cu doping, nor does it compare these values with similar studies on doped YMnO₃.

Comments on the Quality of English Language

It is recommend thoroughly reviewing the text to correct grammatical errors and enhance sentence structure for clarity and readability. 

Author Response

(The authors gave the same response as above.)

Round 2

Reviewer 1 Report

Comments and Suggestions for Authors

The authors present in the revised version the modulation of the structure, magnetic, and dielectric properties of YMnO3 by Cu doping. The authors did not accept most of my previous comments (some of my comments were marked as addressed in blue text, but in general, they did not make any changes or significant modifications). Please, do not send me the same paper without any significant modifications (as I mentioned before), because in this form I suggest rejecting it.

Author Response

Dear Reviewer:

We thank you for giving us the opportunity to revise and resubmit our manuscript materials-3032889 entitled “Modulation of the structure, magnetic and dielectric properties of YMnO3 by Cu doping”. We are thankful to you for pointing out some important modifications. We have thoughtfully taken into account these comments. A thorough revision is in blue throughout the manuscript to reflect the improvement of scientific accuracy and presentation clarity. We hope that all these changes fulfil the requirements to make the manuscript acceptable for publication in materials.

Our collective point-by-point responses to reviewers’ comments are provided in in the following pages. My co-authors and I want to thank you again for giving us the opportunity to revise and resubmit our manuscript to materials

Sincerely yours,

Feng Wan, Ph. D

College of Physics & Electronic Engineering

Xianyang Normal University

Xiyang, Shaanxi 712000, China

Reviewer 2 Report

Comments and Suggestions for Authors

Thank you to the authors for considering most of the comments and suggestions. All the points have been addressed or clarified. I recommend the publication of the revised version of the manuscript. 

Author Response

(The authors gave the same response as above.)
